# Mediation Analysis of Waist Circumference in the Association of Gut Microbiota with Insulin Resistance in Children

**DOI:** 10.3390/children10081382

**Published:** 2023-08-14

**Authors:** Juan Carlos Ayala-García, Cinthya Estefhany Díaz-Benítez, Alfredo Lagunas-Martínez, Yaneth Citlalli Orbe-Orihuela, Ana Cristina Castañeda-Márquez, Eduardo Ortiz-Panozo, Víctor Hugo Bermúdez-Morales, Miguel Cruz, Ana Isabel Burguete-García

**Affiliations:** 1Centro de Investigación Sobre Enfermedades Infecciosas, Instituto Nacional de Salud Pública, Cuernavaca 62100, Mexico; carlos.ayala@insp.edu.mx (J.C.A.-G.); cediaz@insp.mx (C.E.D.-B.); alagunas@insp.mx (A.L.-M.); yaneth.orbe@insp.mx (Y.C.O.-O.); vbermudez@insp.mx (V.H.B.-M.); 2Instituto de Investigación Científica, Universidad Juárez del Estado de Durango, Durango 34000, Mexico; cristy_acm@hotmail.com; 3Centro de Investigación en Salud Poblacional, Instituto Nacional de Salud Pública, Cuernavaca 62100, Mexico; eduardo.ortiz@insp.mx; 4Unidad de Investigación Médica en Bioquímica, Centro Médico Nacional Siglo XXI, Ciudad de México 06720, Mexico; mcruzl@yahoo.com

**Keywords:** gut microbiota, dysbiosis, insulin resistance, overweight, mediation analysis

## Abstract

Background: Persistent gut microbiota (GM) imbalance has been associated with metabolic disease development. This study evaluated the mediating role of waist circumference in the association between GM and insulin resistance (IR) in children. Methods: This cross-sectional study included 533 children aged between 6 and 12. The anthropometry, metabolic markers, and relative abundance (RA) of five intestinal bacterial species were measured. Path coefficients were estimated using path analysis to assess direct, indirect (mediated by waist circumference), and total effects on the association between GM and IR. Results: The results indicated a positive association mediated by waist circumference between the medium and high RA of *S. aureus* with homeostatic model assessments for insulin resistance (HOMA-IR) and for insulin resistance adiponectin-corrected (HOMA-AD). We found a negative association mediated by waist circumference between the low and medium RA of *A. muciniphila* and HOMA-IR and HOMA-AD. Finally, when we evaluated the joint effect of *S. aureus*, *L. casei*, and *A. muciniphila*, we found a waist circumference-mediated negative association with HOMA-IR and HOMA-AD. Conclusions: Waist circumference is a crucial mediator in the association between *S. aureus* and *A. muciniphila* RA and changes in HOMA-IR and HOMA-AD scores in children.

## 1. Introduction

Insulin resistance (IR) is defined as an increase in the production and secretion of insulin in the beta cells of the pancreas as a compensatory process for the alteration in the elimination of glucose in the target tissues—mainly the liver, muscles, and adipose tissue [1]. The childhood obesity epidemic worldwide has directly impacted the increase in IR in this population. The factors associated with the development of IR in children include sex, ethnic origin, chronic low-grade inflammation, cellular dysfunction, and mainly excess visceral adipose tissue [2,3].

During the last decade, the role of gut microbiota (GM) in developing metabolic diseases has been studied. GM has evolved with the human being, generating a symbiotic relationship that allows the host to perform essential physiological functions such as protection against pathogens, development of the immune response, participation in metabolism, digestion, and neuronal development [4,5].

GM is transferred from the mother to the fetus during the birthing process, depending on the route of birth, and colonization continues during the first few years of life. The composition of GM is largely dependent on dietary and environmental factors. However, antibiotic use, stress, and breastfeeding can disturb the composition of the microbiota, altering the physiological functions of the host [6,7]. This alteration in the composition of GM, known as gut dysbiosis, produces different activations of signaling pathways and phenotypic changes in humans, triggering chronic inflammatory processes and cardiometabolic diseases such as obesity, type 2 diabetes (T2D), and cardiovascular diseases [8]. We aimed to evaluate the direct, indirect (mediated by waist circumference), and total effects on the association between GM and IR in Mexican children.

In the international literature, some species such as Akkermansia muciniphila (*A. muciniphila*), Lactobacillus casei (*L. casei*), Lactobacillus paracasei (*L. paracasei*), Lactobacillus reuteri (*L. reuteri*), and Staphylococcus aureus (*S. aureus*) have attracted considerable attention due to their association with the development of metabolic diseases. *A. muciniphila* has been proposed as a potential probiotic in animal and human models; daily supplementation decreases total body weight, fat mass, waist circumference, and metabolic endotoxemia; it inhibits proinflammatory pathways and improves glucose tolerance [9,10].

Some *Lactobacillus* species are associated with protection, while others increase the risk of developing IR; this is due to differences in carbohydrate metabolism. While some species, such as *L. casei* and *L. paracasei*, can store carbohydrates as glycogen because they encode genes for glucose permease, *L. reuteri* lacks these enzymes involved in fructose catabolism [11,12]. Supplementation with *L. casei* and *L. paracasei* affects sirtuin 1 (SIRT 1) and fetuin-A levels, decreases blood glucose and insulin levels, and reduces inflammatory status in subjects with T2D [13,14].

*S. aureus* belongs to the phylum *Firmicutes*; these bacteria produce enzymes involved in the energetic extraction of food and its deposit in fat reserves. Overweight children present a higher abundance of *S. aureus* than normal-weight children, and *S. aureus* produces superantigens (Sags) that induce the production of proinflammatory cytokines in adipocytes, contributing to the development of chronic low-grade inflammation, which plays an essential role in the development of peripheral IR [15,16,17].

Adipose tissue plays an essential role in the effect of gut bacteria on insulin action, secreting several adipokines that participate in metabolic regulation, inflammation, immune function, and processes related to cardiometabolic diseases [18]. Compared to the main adipokines, adiponectin has an inverse relationship with obesity; it has been described that the reduction of body fat is associated with an increase in circulating levels of adiponectin [19]. It is also an insulin sensitizer, promoting insulin action by improving insulin resistance, and it increases the secretion of anti-inflammatory cytokines [20]. Because of its potential benefits, evaluating its role in the development of IR is relevant.

GM participates in the health–disease process of its host, so the present work aimed to evaluate the mediating role of waist circumference in the association between *A. muciniphila*, *L. casei*, *L. paracasei*, *L.reuteri*, and *S. aureus* with homeostatic model assessment for insulin resistance (HOMA-IR) and homeostatic model assessment for insulin resistance adiponectin-corrected (HOMA-AD) in Mexican children.

## 2. Materials and Methods

### 2.1. Design and Study Population

This cross-sectional study was conducted between 2012 and 2014 and approved by the Ethics, Research, and Biosafety Commissions of the Instituto Nacional de Salud Pública (INSP) with the approval number CI:1129, No. 1294. The analysis included the information of 533 unrelated children from 6 to 12 years old, living in four geographic zones of Mexico City (north, south, east, and west). Children diagnosed with infectious diseases or gastrointestinal disorders and those undergoing antibiotic treatment two months before the study were excluded. The data were obtained from previous work [21].

All children and parents signed their assent and informed consent, respectively. We performed the power calculation to detect indirect effects using the Monte Carlo method with an application in the R statistical language. We used the methodology described by Alexander M. Schoemann; using our sample size of n = 533 with 5000 simulations, 20,000 repetitions, and a confidence level of 95%, we obtained a statistical power of 1 [22].

### 2.2. Gut Microbiota

DNA was extracted from a 200 mg stool sample using the commercial kit QIAamp^®^ DNA stool (Qiagen, Hilden, Germany). The DNA concentration and purity was determined using a spectrophotometer NanoDropTM Lite Thermo Scientific^TM^ (Madison, Wisconsin, U.S.A.) with an absorbance of 260 nm and 260/280 nm. The relative abundance (RA) of *A. muciniphila*, *L. casei*, *L. paracasei*, *L. reuteri*, and *S. aureus*, was determined via quantitative polymerase chain reaction (qPCR) using specific universal primers (Appendix A).

Each qPCR reaction was performed in duplicate using 5 μL of 2× Maxima SYBR Green/ROX qPCR Master Mix Thermo Scientific^TM^ (Carlsbad, California, U.S.A.), 1 μL (five pmol) from each primer, 5 ng (*S. aureus*) and 10 ng (rest of species) of the DNA template, and 2 μL of nucleic acid-free water (Fermentas^®^, Carlsbad, California, U.S.A.), resulting in a final volume of 10 μL. The amplifications were performed with the StepOnePlusTM Real-Time PCR System (Applied Biosystems, Woodlands, Singapore, Singapore) under the following conditions: initial thermal cycling of ten minutes at 95 °C, 40 cycles with a denaturation phase at 95 °C for 15 s, an alignment phase at 56–60 °C for 20 s, and an elongation phase at 72 °C for 20 s.

The RA obtained with the following formula was calculated: UAR = 2^−ΔCt^, where UAR = Units of Relative Abundance and ΔCt = Ct specific primer-Ct universal primer [23].

### 2.3. Waist Circumference and Body Mass Index

The personnel who performed the measurement were trained and standardized in the correct measurement of children with calibrated instruments. Waist circumference (cm) was measured using the lower edge of the last palpable rib and the upper edge of the iliac crest around the waist as equidistant reference points. The weight (kg), height (cm), and body mass index (BMI, kg/m^2^) were measured. The children were tested without shoes on and with as little clothing as possible.

### 2.4. Biochemical Determination

Blood samples were collected via antecubital venipuncture under overnight fasting conditions for 12 h. The samples were centrifuged to separate the serum and stored at −80 °C until use.

Serum levels of glucose (mg/dL), insulin (mU/L), total cholesterol (mg/dL), high-density lipoproteins (HDL, mg/dL), low-density lipoproteins (LDL, mg/dL), and triglycerides (mg/dL) were measured via chemiluminescence to evaluate the metabolic status.

Serum adiponectin (µg/mL) concentrations were measured using commercial ELISA kits according to the manufacturer (PeproTech, Rocky Hill, New Jersey, U.S.A.), and absorbance was determined using a Labsystems Multiskan MS^®^, Vantaa, Finland.

### 2.5. Insulin Resistance

We converted glucose units from mg/dL to mmol/L to calculate the HOMA-IR and HOMA-AD, using the methodology of Sullara Vilela, as follows:

HOMA-IR = [glucose (mmol/L) × insulin (mU/L)]/22.5 and HOMA-AD = [glucose (mmol/L) × insulin (mU/L)]/[22.5 × adiponectin (µg/mL)] [24].

### 2.6. Hereditary Family History and Sociodemographic Data

Trained personnel used a questionnaire to collect information from the children and their parents or guardians regarding family history of T2D, high blood pressure, and overweight/obesity, as well as sociodemographic, socioeconomic, and pathological data.

### 2.7. Physical Activity

To collect information on physical activity (METs/hour/week) and inactivity, we used a validated questionnaire for Mexican students (CAINM) consisting of 40 questions [25].

### 2.8. Diet

Dietary intake was obtained through a validated semi-quantitative questionnaire focused on food consumption frequency. For more precise measurements, the questionnaire was administered to the children in the presence of their parents or guardians. The questionnaire included questions about the frequency and portions of 111 foods divided into sections. According to the reported frequency, grams of lipids, proteins, and carbohydrates were calculated as the average daily consumption.

### 2.9. Statistical Analysis

To describe the relevant variables in the population, we stratified children into two groups: normal weight and overweight/obese (OW/OB), according to BMI Z-scores for age [26]. We performed the Shapiro–Wilk test to evaluate normality. As the distribution of the variables was non-normal, between-group comparison was performed using the Mann–Whitney U test for continuous variables and chi-square (X^2^) for categorical variables. For the inferential analysis, we obtained the tertiles of RA for each bacterial species and dichotomized the waist circumference variable above and below the median of our sample (63.6 cm).

We used a logistic regression model to evaluate the association between GM and waist circumference and evaluated the association between GM and waist circumference with IR using linear regression. To consider mediation, we obtained the direct, indirect, and total effects via path analysis, taking the RA tertiles of each bacterial species as the independent variable, waist circumference as the mediator, and the HOMA-IR and HOMA-AD indexes as the dependent variables.

Next, we built a latent variable with the tertiles of the RA of the bacterial species in which we found an association with HOMA-IR and HOMA-AD; we used this latent variable as the independent variable, waist circumference as a mediator, and IR indexes as dependent variables. Figure 1 shows the Directed Acyclic Diagram (DAG), in which we can observe the causal relationship between GM and IR mediated by waist circumference. The confounding adjustment set included a family history of T2D, a family history of overweight/obesity (OW/OB), age, physical activity, sex, diet, and U (unmeasured) genes. However, the history of T2D and physical activity did not add variability to our estimator, so we decided not to include them as adjustment variables in order to ensure we had the most parsimonious models possible. We established statistical significance with a value of *p* < 0.05. All analyses were performed using the Stata® software version 17.

## 3. Results

We evaluated the general characteristics of 533 children according to the BMI Z-scores. Table 1 shows the median, 25th percentile, and 75th percentile for continuous variables and proportions for categorical variables.

The combined prevalence of (OW/OB) was 49%. Children with OW/OB had a higher median age, a higher prevalence of a history of OW/OB, a higher prevalence of a history of T2D, a greater waist circumference, and a greater median in the score of the HOMA-IR and HOMA-AD indexes compared to normal-weight children. We found no difference between the two groups in the distribution by physical activity, sex, macronutrient consumption, or RA of bacterial species.

The prevalence of children with HOMA-IR > 2 was 7% (n = 38), and the prevalence of children with HOMA-IR < 2 was 93% (n = 495).

When evaluating the metabolic status of both groups, we found that children with OW/OB had higher serum levels of cholesterol, triglycerides, LDL, and insulin and lower serum levels of HDL than normal-weight children (Appendix A).

We analyzed the association between the RA of bacterial species and waist circumference, and as shown in Table 2, children with medium and high RA of *S. aureus* had higher odds of having a higher waist circumference than children with low RA of *S. aureus* [OR 2.30 (95% CI: 1.44, 3.65) and OR 1.72 (95% CI: 1.08, 2.73)]. Children with medium RA of *L. casei* had lower odds of having a higher waist circumference than children with high RA of *L. casei* [OR 0.63 (95% CI: 0.40, 1.00)]. Finally, children with low and medium RA of *A. muciniphila* had lower odds of having a higher waist circumference than children with high RA of *A. muciniphila* [OR 0.62 (95% CI: 0.39, 0.99) and OR 0.50 (95% CI: 0.32, 0.80)].

Through linear regression, we evaluated the association between the tertiles of RA of each bacterial species and waist circumference with IR indexes. After adjusting for confounders, we determined that children with higher waist circumferences showed a 62% increase in their HOMA-IR score [β = 0.62 (95% CI: 0.49, 0.75)] and a 16% increase in their HOMA-AD score [β = 0.16 (95% CI: 0.13, 0.20)] than children with lower waist circumferences (Table 3).

Table 4 shows the direct, indirect (mediated by waist circumference), and total effects obtained via path analysis (Appendix A). Children with medium RA of *L. paracasei* showed a 15% increase in their HOMA-IR score compared to children with high RA of *L. paracasei* [PC = 0.15 (95% CI: 0.002, 0.30)].

Regarding indirect effects, the waist circumference measurements show that children with medium RA of *S. aureus* had an 11% increase in HOMA-IR score [PC = 0.11 (95% CI: 0.04, 0.17)] and a 3% increase in HOMA-AD score [PC = 0.03 (95% CI: 0.01, 0.05)] compared to children with low RA of *S. aureus*. The results also showed that children with high RA of *S. aureus* exhibited a 7% increase in HOMA-IR score [PC = 0.07 (95% CI: 0.01, 0.13)] and a 2% increase in HOMA-AD score [PC = 0.02 (95% CI: 0.002, 0.3)] compared to children with low RA of *S. aureus*.

We found that children with low RA of *A. muciniphila* had a 6% decrease in HOMA-IR score [PC = −0.06 (95% CI: −0.13, −0.0013)] and a 2% decrease in HOMA-AD score [PC= -0.02 (95% CI: −0.03, −0.0004)] compared to children with high RA of *A. muciniphila*. Finally, children with medium RA of *A. muciniphila* had a 9% decrease in HOMA-IR score [PC= −0.09 (95% CI: −0.16, −0.03)] and a 2% decrease in HOMA-AD score [PC= −0.02 (95% CI: −0.04, −0.007)] compared to children with high RA of *A. muciniphila*.

To evaluate the joint effect of the bacterial species under study, we constructed a latent variable (microbiota) with the observed RA of *S. aureus*, *A. muciniphila*, and *L. casei*.

We performed structural equation modeling to evaluate the direct, indirect (mediated by waist circumference), and total effects of the latent variable microbiota with HOMA-IR and HOMA-AD (Appendix A). We found that waist circumference is a mediator in children with a profile characterized by high RA of *S. aureus*, medium RA of *L. casei*, and low RA of *A. muciniphila*; there was a 63% increase in HOMA-IR score [PC = 0.63 (95% CI: 0.50, 0.78)] and a 17% increase in HOMA-AD score [PC = 0.17 (95% CI: 0.13, 0.20)].

Finally, based on waist circumference measurements in children with a profile characterized by high RA of *S. aureus*, medium RA of *L. casei*, and medium RA of *A. muciniphila*, we identified a 66% increase in HOMA-IR score [PC = 0.66 (95% CI: 0.51, 0.81)] and a 17% increase in HOMA-AD score [PC = 0.17 (95% CI: 0.13, 0.20)] (Table 5).

## 4. Discussion

In our study, *S. aureus* was indirectly associated with increased scores for the HOMA-IR and HOMA-AD indexes. A study conducted in mice showed that *S. aureus* impairs glucose tolerance through the secretion of an extracellular domain of insulin-binding proteins that block insulin-mediated glucose uptake [27]. Our research revealed that *S. aureus* is associated with increased waist circumference, triglyceride and LDL levels, and lower HDL levels [28].

In prospective studies, the results of the characterization of the intestinal microbiota in women according to their body mass index (BMI) before pregnancy showed that pregnant women who were overweight had a higher RA of *S. aureus* than normal-weight women. After delivery, the composition of the fecal microbiota in the infants was analyzed. The children of women who were overweight before pregnancy had a higher RA of *S. aureus* than the children of normal-weight women, which implies a direct transmission of the maternal microbiota to the child depending on the degree of adiposity [29,30]. Our results were concordant since we did not find a direct association between *S. aureus*, HOMA-IR, and HOMA-AD. The association was observed only when we included waist circumference as a mediator.

We also found that *A. muciniphila* was indirectly associated with decreased HOMA-IR and HOMA-AD scores. A study carried out in mice revealed that 5 weeks of supplementation with *A. muciniphila* reduced visceral fat, which is closely related to the pathogenesis of IR [31]. In a cohort study in the Chinese population, it was reported that a decrease in the abundance of *A. muciniphila* was associated with impaired IR in subjects with T2D because *A. muciniphila* modulates insulin secretion through the levels of 3 *β*—serum chenodeoxycholic acid, inducing glycogen synthesis and suppressing gluconeogenesis, thereby improving glucose tolerance [32].

Our findings are similar to those of other authors; in overweight and obese children, a decrease in the abundance of *A. muciniphila* was found compared to that of normal-weight children [33]. *A. muciniphila* can restore intestinal barrier function and proper expression of tight junctions, causing thickening of the intestinal mucosal layer, decreasing chronic low-grade inflammation, and improving the metabolic health of the host [34].

We identified an association between GM and IR mediated by waist circumference, as GM is an ecosystem in equilibrium that can be altered to modify the relative abundances of bacteria compared to their normal abundance; this persistent imbalance of the microbiota is known as dysbiosis [35].

Intestinal dysbiosis facilitates the disorganization of the tight junction proteins of the colonic epithelial cells ZO-1 and occludins and a reduction in the intestinal mucus layer, favoring intestinal permeability and a more significant endotoxin load. Under these conditions, lipopolysaccharide (LPS) from the outer membrane of Gram-negative bacteria can translocate through the epithelial layers into the bloodstream, activating toll-like receptor (TLR) 4 and thus inducing the production of proinflammatory cytokines and interferons [36,37,38,39].

In addition to the inflammation caused by LPS, obese individuals have hypertrophic adipocytes and a more significant number of macrophages; because of adipocyte hypertrophy, the blood supply is compromised, causing oxygen deficiency and macrophage infiltration towards adipose tissue, inducing the overexpression of proinflammatory cytokines, and reducing the expression of adiponectin [40,41]. Therefore, excess abdominal adiposity would potentiate the effect of LPS-induced endotoxemia and the subsequent metabolic effects of low-grade inflammation, such as IR and dyslipidemia [42].

When evaluating the joint effect of bacterial profiles, we observed a dominant effect of *S. aureus* over *L. casei* and *A. muciniphila*; even the high relative abundance of the latter does not offset the increase in HOMA-IR and HOMA-AD scores. Therefore, studies must evaluate how *S. aureus* inhibits the protective effect of *L. casei* and *A. muciniphila* on RI.

Regarding HOMA-IR and HOMA-AD, we noted no difference between using one index or the other; some studies have reported similar results in which children with metabolic syndrome have higher scores in both indexes than healthy children. Additionally, they have achieved efficient screening for metabolic risk using both indexes without observing significant differences [43,44]. When adjusting the HOMA index for adiponectin levels, slightly more accurate estimates are observed than those obtained using HOMA-IR, but there is no difference in the direction or magnitude of the association.

Within the limitations of our study, we evaluated RI using the HOMA index, with the awareness that the hyperinsulinemic–euglycemic clamp (HEC) is the gold standard for assessing insulin sensitivity. The HOMA index imputes the dynamic function of β-cells, which prevents direct measurement of these cells’ proper insulin secretion function. Nevertheless, the HOMA index is an inexpensive quantitative tool that works adequately in population studies. The validity between the HOMA index and the HEC in the pediatric population has also been reported, allowing us to use this index as a valuable tool for classification in epidemiological studies [45,46,47].

Among anthropometric markers of central adiposity, waist circumference is the best predictor of insulin resistance in children. Although dual-energy X-ray absorptiometry (DXA) is the gold standard for measuring body composition, waist circumference has been chosen for its simple, fast, and cheap measurement in population-based studies. Several studies have reported a high correlation between waist circumference and metabolic diseases. During the last few years, waist circumference has been encouraged in clinical practice to detect subjects at increased risk of metabolic diseases [48]. If there were a measurement error among the anthropometrists, it would be non-differential.

Finally, due to the nature of our design, we consider the probability of reverse causality; however, our analysis strategy (path analysis and structural equation modeling) allows us to evaluate the fit of theoretical models in which a set of dependency relationships between variables is proposed. These methods do not prove causality but allow selection or inference between causal hypotheses [49]. In addition, in prospective studies, it has been determined that intestinal dysbiosis leads to an increased risk of developing IR [50].

The results allow us to consider GM as a potentially useful biomarker for identifying children at risk of developing IR and, later, T2D, considering the effect of waist circumference. Measuring the RA of intestinal bacteria will help to carry out interventions for at-risk children in a timely and effective manner to reduce the prevalence of IR in the child population, directly impacting their quality of life.

## 5. Conclusions

Waist circumference mediates the association between the RA of *S. aureus* and *A. muciniphila* and insulin resistance. When evaluating microbiota profiles’ joint effect, we observed that *S. aureus* predominates over *L. paracasei* and *A. muciniphila*, increasing HOMA-IR and HOMA-AD scores. However, more studies are needed to elucidate the mechanism by which the gut microbiota is associated with insulin resistance in children, allowing potential areas of opportunity in the treatment and approach to the care of children with abdominal obesity.

## Figures and Tables

**Figure 1 children-10-01382-f001:**
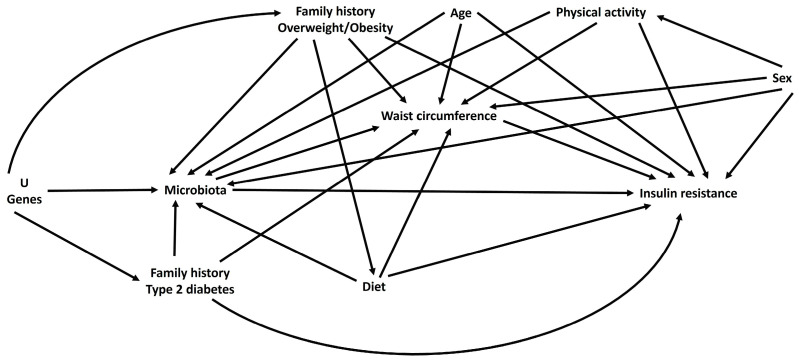
Directed acyclic graph (DAG) representing the hypothetical causal structure.

**Table 1 children-10-01382-t001:** General characteristics of participants according to BMI Z-scores.

Characteristics ^a^ n = 533	Normal Weight	Overweight/Obesity	*p*-Value
	n = 265 (51%)	n = 268 (49%)	
Age (years)	9 (7–10)	9 (8–10)	**0.013**
Physical activity (METs/hour/week)	301 (150–577)	337 (162–625)	0.330
Sex			
Boy (%)	52	48	0.368
Girl (%)	48	52	
Family History of OW/OB			
Yes (%)	43	57	**0.001**
No (%)	58	42	
Family History of T2D			
Yes (%)	37	63	**0.020**
No (%)	52	48	
Macronutrient Consumption			
Carbohydrates (g/day)	272.01 (210.27–339.53)	272.64 (211.07–357.14)	0.571
Lipids (g/day)	74.84 (59.28–93.39)	74.81 (58.18–93.66)	0.989
Proteins (g/day)	69.66 (55.15–83.43)	68.13 (53.48–86.88)	0.963
Exposure Variables			
*S. aureus* (RA)	0.0000368 (7.54 × 10^−6^–0.0002222)	0.0000558 (9.83 × 10^−6^–0.000277)	0.140
*L. paracasei* (RA)	0.0002953 (0.0000239–0.0025059)	0.0003333 (0.0000212–0.0044035)	0.622
*L. casei* (RA)	0.0004417 (0.0000628–0.0064059)	0.0004487 (0.0000457–0.0075155)	0.858
*L. reuteri* (RA)	0.0001371 (0.0000198–0.0009797)	0.0001141 (0.0000202–0.0007388)	0.521
*A. muciniphila* (RA)	0.0033518 (0.0000323–0.0585953)	0.0034792 (0.0000413–0.0742626)	0.539
Mediator Variable			
Waist circumference (cm)	56.7 (53.5–61.2)	74.4 (67.7–79.9)	**<0.001**
Outcome Variables			
HOMA-IR	0.36 (0.28–0.53)	0.81 (0.41–1.55)	**<0.001**
HOMA-AD	0.06 (0.05–0.10)	0.17 (0.06–0.21)	**<0.001**

^a^ Values represented the median (p25 and p75) or percentages. The Mann–Whitney U test is used for continuous variables and the chi-square for categorical variables. OW/OB: overweight/obesity; T2D: type 2 diabetes; RA: relative abundance; HOMA-IR: homeostatic model assessment for insulin resistance; HOMA-AD: homeostatic model assessment for insulin resistance adiponectin-corrected. Statistically significant differences are marked in bold: *p* < 0.05.

**Table 2 children-10-01382-t002:** Association between gut microbiota and waist circumference.

	Waist Circumference ≥ 63.6 cm		
	OR	95% CI	*p*-Value
RA of *Staphylococcus aureus **			
Medium tertile	2.30	1.44, 3.65	**<0.001**
High tertile	1.72	1.08, 2.73	**0.022**
RA of *Lactobacillus reuteri* *			
Medium tertile	1.19	0.76, 1.89	0.447
High tertile	1.21	0.77, 1.90	0.417
RA of *Lactobacillus paracasei ***			
Low tertile	0.82	0.52, 1.29	0.393
Medium tertile	0.89	0.57, 1.41	0.639
RA of *Lactobacillus casei ***			
Low tertile	0.78	0.49, 1.23	0.283
Medium tertile	0.63	0.40, 1.00	**0.050**
RA of *Akkermansia muciniphila ***			
Low tertile	0.62	0.39, 0.99	**0.046**
Medium tertile	0.50	0.32, 0.80	**0.004**

RA: relative abundance. Reference group: * Low tertile, ** High tertile. Logistic regression adjusted by age, sex, family history of OW/OB, and daily consumption of carbohydrates, lipids, and proteins. Statistically significant differences are marked in bold: *p* < 0.05.

**Table 3 children-10-01382-t003:** Association between waist circumference and gut microbiota with HOMA-IR and HOMA-AD.

	HOMA-IR			HOMA-AD		
	β	95% CI	*p*-Value	β	95% CI	*p*-Value
Waist circumference ≥ 63.6 cm *	0.62	0.49, 0.75	**<0.001**	0.16	0.13, 0.20	**<0.001**
RA of *Staphylococcus aureus ***						
Medium tertile	0.15	−0.01, 0.31	0.066	0.03	−0.01, 0.07	0.111
High tertile	0.04	−0.12, 0.20	0.616	0.01	−0.03, 0.06	0.459
RA of *Lactobacillus reuteri* **						
Medium tertile	0.05	−0.11, 0.21	0.544	0.006	−0.03, 0.05	0.766
High tertile	−0.08	−0.24, 0.08	0.345	−0.005	−0.05, 0.04	0.801
RA of *Lactobacillus paracasei ****						
Low tertile	−0.10	−0.27, 0.06	0.210	−0.02	−0.06, 0.02	0.407
Medium tertile	0.13	−0.03, 0.29	0.105	0.03	−0.01, 0.07	0.141
RA of *Lactobacillus casei ****						
Low tertile	−0.06	−0.22, 0.11	0.492	−0.01	−0.05, 0.03	0.585
Medium tertile	−0.07	−0.23, 0.10	0.445	−0.02	−0.06, 0.02	0.289
RA of *Akkermansia muciniphila ****						
Low tertile	0.04	−0.12, 0.21	0.611	0.02	−0.02, 0.06	0.410
Medium tertile	0.05	−0.11, 0.21	0.527	0.002	−0.04, 0.04	0.907

RA: relative abundance. Reference group: * Waist circumference <63.6 cm; ** Low tertile; *** High tertile. Linear regression adjusted by age, sex, family history of OW/OB, and daily consumption of carbohydrates, lipids, and proteins. Statistically significant differences are marked in bold: *p* < 0.05.

**Table 4 children-10-01382-t004:** Direct, indirect, and total effects of gut microbiota on HOMA-IR and HOMA-AD.

	HOMA-IR			HOMA-AD		
	Path Coefficient (PC)	95% CI	*p*-Value	Path Coefficient (PC)	95% CI	*p*-Value
**Direct effect**						
RA of *Staphylococcus aureus **						
Medium tertile	0.043	−0.11, 0.19	0.574	0.0048	−0.03, 0.04	0.807
High tertile	−0.03	−0.18, 0.12	0.696	−0.0033	−0.04, 0.03	0.868
RA of *Lactobacillus reuteri* *						
Medium tertile	0.025	−0.12, 0.17	0.744	−0.00042	−0.04, 0.04	0.983
High tertile	−0.11	−0.25, 0.04	0.165	−0.013	−0.05, 0.02	0.518
RA of *Lactobacillus paracasei ***						
Low tertile	−0.076	−0.22, 0.07	0.317	−0.01	−0.04, 0.03	0.598
Medium tertile	0.15	0.002, 0.30	**0.047**	0.036	−0.002, 0.07	0.066
RA of *Lactobacillus casei ***						
Low tertile	−0.036	−0.19, 0.11	0.639	−0.0023	−0.04, 0.04	0.906
Medium tertile	−0.0016	−0.15, 0.15	0.983	−0.0063	−0.04, 0.03	0.748
RA of *Akkermansia muciniphila ***						
Low tertile	0.11	−0.04, 0.26	0.158	0.035	−0.003, 0.07	0.074
Medium tertile	0.15	−0.003, 0.30	0.056	0.027	−0.01, 0.06	0.164
**Indirect effect**						
RA of *Staphylococcus aureus **						
Medium tertile	0.11	0.04, 0.17	**0.001**	0.03	0.01, 0.05	**0.001**
High tertile	0.07	0.01, 0.13	**0.024**	0.02	0.002, 0.03	**0.024**
RA of *Lactobacillus reuteri* *						
Medium tertile	0.02	−0.04, 0.09	0.417	0.007	−0.009, 0.02	0.417
High tertile	0.03	−0.03, 0.09	0.387	0.007	−0.01, 0.02	0.386
RA of *Lactobacillus paracasei ***						
Low tertile	−0.03	−0.09, 0.03	0.373	−0.007	−0.02, 0.009	0.373
Medium tertile	−0.02	−0.08, 0.04	0.610	−0.004	−0.02, 0.01	0.610
RA of *Lactobacillus casei ***						
Low tertile	−0.03	−0.10, 0.03	0.260	−0.009	−0.03, 0.007	0.260
Medium tertile	−0.06	−0.01, 0.0006	0.052	−0.02	−0.03, 0.0001	0.052
RA of *Akkermansia muciniphila ***						
Low tertile	−0.06	−0.13, −0.0013	**0.045**	−0.02	−0.03, −0.0004	**0.045**
Medium tertile	−0.09	−0.16, −0.03	**0.005**	−0.02	−0.04, −0.007	**0.005**
**Total effect**						
RA of *Staphylococcus aureus **						
Medium tertile	0.15	−0.01, 0.31	0.063	0.035	−0.007, 0.07	0.108
High tertile	0.04	−0.12, 0.20	0.612	0.017	−0.02, 0.06	0.455
RA of *Lactobacillus reuteri* *						
Medium tertile	0.05	−0.11, 0.21	0.514	0.006	−0.03, 0.05	0.764
High tertile	−0.08	−0.24, 0.08	0.385	−0.005	−0.05, 0.04	0.799
RA of *Lactobacillus paracasei ***						
Low tertile	−0.10	−0.26, 0.05	0.205	−0.02	−0.06, 0.02	0.402
Medium tertile	0.13	−0.03, 0.29	0.101	0.03	−0.01, 0.07	0.136
RA of *Lactobacillus casei ***						
Low tertile	−0.07	−0.24, 0.09	0.387	−0.01	−0.05, 0.03	0.582
Medium tertile	−0.06	−0.22, 0.10	0.441	−0.02	−0.06, 0.02	0.284
RA of *Akkermansia muciniphila ***						
Low tertile	0.04	−0.12, 0.20	0.607	0.02	−0.02, 0.06	0.406
Medium tertile	0.05	−0.11, 0.21	0.523	0.002	−0.04, 0.04	0.906

RA: relative abundance. Reference group: * Low tertile, ** High tertile. Path analysis was adjusted by age, sex, family history of OW/OB, and daily consumption of carbohydrates, lipids, and proteins. Statistically significant differences are marked in bold: *p* < 0.05.

**Table 5 children-10-01382-t005:** Direct, indirect, and total effects of gut microbiota profiles on HOMA-IR and HOMA-AD.

	HOMA-IR			HOMA-AD		
	Path Coefficient (PC)	95% CI	*p*-Value	Path Coefficient (PC)	95% CI	*p*-Value
**Direct effect**						
Gut microbiota						
Profile 1	−1.2	−4.6, 2.1	0.479	−0.4	−1.5, 0.76	0.500
Profile 2	−0.98	−2.45, 0.49	0.190	−0.092	−0.42, 0.24	0.588
**Indirect effect**						
Gut microbiota						
Profile 1	0.63	0.50, 0.78	**<0.001**	0.17	0.13, 0.20	**<0.001**
Profile 2	0.66	0.51, 0.81	**<0.001**	0.17	0.13, 0.20	**<0.001**
**Total effect**						
Gut microbiota						
Profile 1	−0.57	−3.9, 2.8	0.736	−0.23	−1.38, 0.93	0.699
Profile 2	−0.32	−1.76, 1.12	0.664	0.07	−0.24, 0.40	0.642

Profile 1: high RA of *S. aureus*, medium RA of *L. casei*, and low RA of *A. muciniphila*. Profile 2: high RA of *S. aureus*, medium RA of *L. casei*, and medium RA of *A. muciniphila*. Structural equation modeling is adjusted by age, sex, family history of OW/OB, and daily consumption of carbohydrates, lipids, and proteins. Statistically significant differences are marked in bold: *p* < 0.05.

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
