# Peer review of "Mediation Analysis of Waist Circumference in the Association of Gut Microbiota with Insulin Resistance in Children"

_children, 2023, doi:10.3390/children10081382_

Round 1

Reviewer 1 Report (Previous Reviewer 3)

The authors have adrressed my previous comments.

English language requires minor editing.

Author Response

Dear Reviewer 1

We appreciate the time you have taken to review our manuscript; your contributions have improved the quality of our work.

English language editing has been performed, as an attached document you will find the certificate. 

We hope that all questions have been answered successfully.

Reviewer 2 Report (Previous Reviewer 1)

Please specify:

How many children in this research have waist circumference > 63.6cm in the normal weight group and OW/OB group;

Did these group children (normal weight and OW/OB) have HOMA-IR above 2? And how many children?

Please explain the limitations of the study regarding insulin resistance, waist circumference, GM etc. 

Author Response

Dear Reviewer 2

The comments made on our manuscript have substantially improved its quality; we appreciate your comments.

English language editing has been performed, as an attached document you will find the certificate.

Regarding your questions:

  1. How many children in this research have waist circumference > 63.6cm in the normal weight group and OW/OB group?

Our sample is composed as follows:

Normal weight with waist circumference >63.6 cm: 37 children.

OW/OB with waist circumference >63.6 cm: 229 children.

normal weight with waist circumference <63.6 cm: 228 children.

OW/OB with waist circumference <63.6 cm: 39 children.

Total sample: 533 children.

Due to the classification techniques, we can observe discordant pairs (normal weight with waist circumference >63.6 cm and OW/OB with waist circumference <63.6 cm). The BMI Z-score allows us to classify children according to age, sex, height, and weight, but it is not a good indicator of adiposity. Waist circumference is a proxy for abdominal adiposity, highly correlated with insulin resistance. For this reason, we decided to classify the children according to their waist circumference.

  1. Did these group children (normal weight and OW/OB) have HOMA-IR above 2? And how many children?

Regarding the HOMA-IR score, 38 children have HOMA-IR >2, and 495 children have HOMA-IR <2, for a total of 533 children.

We decided to use the HOMA-IR and HOMA-AD index as continuous variables to observe changes in these indices concerning the relative abundances of bacterial species (mediated by waist circumference) because we have a small sample of children with HOMA-IR >2.

  1. Please explain the limitations of the study regarding insulin resistance, waist circumference, GM etc.

The limitations of the use of HOMA can be found on the page 13 lines 355-362.

“We evaluated RI using the HOMA index, with the awareness that the hyperinsulinemic–euglycemic clamp (HEC) is the gold standard for assessing insulin sensitivity. The HOMA index imputes the dynamic function of β-cells, which prevents direct measurement of these cells' proper insulin secretion function. Nevertheless, the HOMA index is an inexpensive quantitative tool that works adequately in population studies. The validity between the HOMA index and the HEC in the pediatric population has also been reported, allowing us to use this index as a valuable tool for classification in epidemiological studies.”

Among anthropometric markers of central adiposity, waist circumference is the best predictor of insulin resistance in children. Although dual-energy X-ray absorptiometry (DXA) is the gold standard for measuring body composition, waist circumference has been chosen for its simple, fast, and cheap measurement in population-based studies. Several studies have reported a high correlation between waist circumference and metabolic diseases. During the last few years, waist circumference has been encouraged in clinical practice* better to detect subjects at increased risk of metabolic diseases. The use of waist circumference is not a limitation in our study; the personnel who performed the measurement were trained and standardized in the correct measurement in children. If there were a measurement error among the anthropometrists, it would be non-differential.

* Ross, R et al (2020). Waist circumference as a vital sign in clinical practice: a Consensus Statement from the IAS and ICCR Working Group on Visceral Obesity. Nature reviews. Endocrinology16(3), 177–189.

Regarding measuring intestinal microbiota, the whole procedure was supervised under strict quality control procedures. The analysts were trained, standardized, and unaware of the study hypothesis; if there were a measurement error among the different analysts, it would be non-differential. Another option to evaluate metabolic endotoxemia is the quantification of lipopolysaccharide. However, it is not a viable option because the blood sample collection tubes may contain exogenous endotoxins that can cause false positives, the assay is sensitive to lipemia, it is not specific for endotoxins from bacteria, and finally, the half-life of lipopolysaccharide is short as it quickly binds to lipopolysaccharide-binding protein (LBP)**. For this reason, quantifying the relative abundance by sequencing is not a limitation.

** Mohammad, S et al (2021). Role of Metabolic Endotoxemia in Systemic Inflammation and Potential Interventions. Frontiers in immunology11, 594150.

Reviewer 3 Report (New Reviewer)

Reviewer Comments

The article addresses two very important and topical issues, obesity in children and gut microbiota. Indeed, gut microbiota has been shown to be relevant in human health and its dysbiosis has been associated with MetS, a health condition linked to the onset of relevant diseases including T2DM.

The research process and the research tool are presented in a logical and professional manner.

In addition to the statistical validity of the research results presented, their practical relevance increases the value of the manuscript.

INTRODUCTION:

-        Adiponectin was used to calculate HOMA-AD and therefore an important factor predicting your results and conclusions. Please add a paragraph of adiponectin in an Introduction and Discussion section.

METHODS:

-        Gut microbiota is shaped by multiple factors such as diet, drug treatments, socioeconomic status,… please clearly describe which were inclusion and exclusion criteria? Did you check also socioeconomic status?

-                Firmicutes, Bacteroidetes, Actinobacteria, Fusobacteria, Proteobacteria, and Verrucomicrobia are the most abundant phyla in gut, with 90% corresponding to Firmicutes and Bacteroidetes. Why didn’t you check/study also some species from the these phylum? It has been shown that obese children and children with MetS have a higher abundance of Firmicutes, Proteobacteria and Actinobacteria and a lower abundance of Bacteroidetes.

-                Since you did not use LDL, HDL, TAG and TC in results it is not important to include them in the manuscript and they can be removed from the Method and Result section.

DISCUSSION AND CONCLUSION

-        Please add a paragraph of adiponectin in a Discussion section.

-        Conclusion is too concrete. It is not just lifestyle that can affect gut microbiota, but also geographic region, and individual variability in the human microbial maturation process must be considered before designing gut microbiome–based interventions to treat or prevent MetS or other associated diseases. Now, knowing that the gut microbiota of children is more flexible, additional efforts must be made to study this population which is prompt to respond positively and rapidly to microbial changes that may reverse deleterious medical conditions before developing irreversible diseases.

Therefore in conclusion the sentence “ More studies are needed to …….”

Author Response

Dear Reviewer 3

We appreciate your suggestions to improve the quality of our manuscript; we look forward to answering all your questions.

English language editing has been performed, as an attached document you will find the certificate.

Below you will find our responses to your questions. In the manuscript, you will find the changes highlight and bold.

  1. Adiponectin was used to calculate HOMA-AD and therefore an important factor predicting your results and conclusions. Please add a paragraph of adiponectin in an Introduction and Discussion section.

The following paragraph has been added on the page 2 lines 78-86.

“Adipose tissue plays an essential role in the effect of gut bacteria on insulin action, secreting several adipokines that participate in metabolic regulation, inflammation, immune function, and processes related to cardiometabolic diseases. Compared to the main adipokines, adiponectin has an inverse relationship with obesity; it has been described that the reduction of body fat is associated with an increase in circulating levels of adiponectin. It is also an insulin sensitizer, promoting insulin action by improving insulin resistance, and it increases the secretion of anti-inflammatory cytokines. Because of its potential benefits, evaluating its role in the development of IR is relevant.”

  1. Gut microbiota is shaped by multiple factors such as diet, drug treatments, socioeconomic status,… please clearly describe which were inclusion and exclusion criteria? Did you check also socioeconomic status?

The selection criteria have been written in more detail on the page 3 lines 101-105.

“The analysis included the information of 533 unrelated children from 6 to 12 years old, living in four geographic zones of Mexico City (North, South, East, and West). Children diagnosed with infectious diseases or gastrointestinal disorders, and those undergoing antibiotic treatment two months before the study, were excluded.”

Socioeconomic status (SES) was evaluated as a potential confounder because it may be associated with gut microbiota and insulin resistance. We found no differences by SES between the different strata of our population. When SES was included as an adjustment variable in the statistical models, it did not provide changes to our estimator, probably because of the homogeneous distribution of this variable. For this reason, we decided not to include SES as a relevant variable for this manuscript.

  1. Firmicutes, Bacteroidetes, Actinobacteria, Fusobacteria, Proteobacteria, and Verrucomicrobia are the most abundant phyla in gut, with 90% corresponding to Firmicutes and Bacteroidetes. Why didn’t you check/study also some species from the these phylum? It has been shown that obese children and children with MetS have a higher abundance of Firmicutes, Proteobacteria and Actinobacteria and a lower abundance of Bacteroidetes.

We agree with the observation. International literature has described that metabolic endotoxemia and metabolic disturbances are mainly caused by an increase in Firmicutes and a decrease in Bacteroidetes. Our initial work*** was oriented to the evaluation of species associated with a higher risk of metabolic alterations (phylum Firmicutes, i.e., Lactobacillus sp and Staphylococcus aureus); however, we measured Akkermansia muciniphila, bacteria that have been evaluated as a potential probiotic with health benefits. We have a large bank of biologicals; our immediate intention is to quantify the relative abundance of other species of interest, such as Bifidobacterium longum, Bifidobacterium adolescentis, Bifidobacterium breve (phylum Actinobacteria) and Bilophilla wadsworthia (phylum Proteobacteria). Unfortunately, we do not have the information at this time.

*** Estrada-Velasco et al (2015). La obesidad infantil como consecuencia de la interacción entre Firmicutes y el consumo de alimentos con alto contenido energético. Nutrición Hospitalaria31(3), 1074-1081.

*** Orbe-Orihuela et al (2018). High relative abundance of and increased TNF-α levels correlate with obesity in children. Salud Pública de México60(1), 5-11. 

*** Huerta-Ávila et al (2019). High relative abundance of Lactobacillus reuteri and fructose intake are associated with adiposity and cardiometabolic risk factors in children from Mexico City. Nutrients11(6), 1207.

*** Castañeda-Márquez et al (2020). Lactobacillus paracasei as a protective factor of obesity induced by an unhealthy diet in children. Obesity research & clinical practice14(3), 271–278.

*** Ayala-García et al (2022). High relative abundance of Staphylococcus aureus and serum cytokines are associated with cardiometabolic abnormalities in children. Metabolic syndrome and related disorders20(5), 303–311.

*** Bahena-Román et al (2022). Low abundance of Akkermansia muciniphila and low consumption of polyphenols associated with metabolic disorders in child population. Human Nutrition and Metabolism, 30(2022), 200167.

  1. Since you did not use LDL, HDL, TAG and TC in results it is not important to include them in the manuscript and they can be removed from the Method and Result section.

We appreciate the observation; however, we consider it relevant to know the metabolic status of the participants. We have included the information on metabolic status as supplementary material.

The following paragraph has been added on the page 6 lines 224-226.

“When evaluating the metabolic status of both groups, we found that children with OW/OB had higher serum levels of cholesterol, triglycerides, LDL, and insulin and lower serum levels of HDL than normal weight children (Table S2).”

5. Please add a paragraph of adiponectin in a Discussion section.

The following paragraph has been included on pages 12 and 13, lines 339-346.

“Regarding HOMA-IR and HOMA-AD, we noted no difference between using one index or the other; some studies have reporte

d similar results in which children with metabolic syndrome have higher scores in both indexes than healthy children. Also, they have achieved efficient screening for metabolic risk using both indexes without observing significant differences. When adjusting the HOMA index for adiponectin levels, slightly more accurate estimates are observed than those obtained using HOMA-IR, but there is no difference in the direction or magnitude of the association.”

6. Conclusion is too concrete. It is not just lifestyle that can affect gut microbiota, but also geographic region, and individual variability in the human microbial maturation process must be considered before designing gut microbiome–based interventions to treat or prevent MetS or other associated diseases. Now, knowing that the gut microbiota of children is more flexible, additional efforts must be made to study this population which is prompt to respond positively and rapidly to microbial changes that may reverse deleterious medical conditions before developing irreversible diseases. Therefore in conclusion the sentence “ More studies are needed to …….”

The correction has been made.

“Waist circumference mediates the association between the RA of S. aureus and A. muciniphila with insulin resistance. When evaluating microbiota profiles' joint effect, we observed that S. aureus predominates over L. paracasei and A. muciniphila, increasing HOMA-IR and HOMA-AD scores. However, more studies are needed to elucidate the mechanism by which the gut microbiota is associated with insulin resistance in children, allowing potential areas of opportunity in the treatment and approach for the care of children with abdominal obesity.”

Round 2

Reviewer 2 Report (Previous Reviewer 1)

1. This citation is from discussion part:

Within the limitations of our study, the food frequency questionnaire (FFQ) is the 347mostfrequentlyused method in population studies as it allows long-term dietary intake 348to be assessed in a relatively simple, cost-effective, and time-efficient way. The diet infor-349mation was obtained throughaself-report. We are aware of participants’ inability to re-350member their intakes accurately and completely; however, we adjusted the models based 351on macronutrient consumption (“all-components model”) to provide a less biased esti-352mate[45,46].In addition, the FFQ was applied in the same way to all the participants in 353the presence of their parents to standardize the information.

Please change this explanation about study limitation, like you answer me in comments:

We evaluated RI using the HOMA index, with the awareness that the hyperinsulinemic–euglycemic clamp (HEC) is the gold standard for assessing insulin sensitivity. The HOMA index imputes the dynamic function of β-cells, which prevents direct measurement of these cells' proper insulin secretion function. Nevertheless, the HOMA index is an inexpensive quantitative tool that works adequately in population studies. The validity between the HOMA index and the HEC in the pediatric population has also been reported, allowing us to use this index as a valuable tool for classification in epidemiological studies.”

Among anthropometric markers of central adiposity, waist circumference is the best predictor of insulin resistance in children. Although dual-energy X-ray absorptiometry (DXA) is the gold standard for measuring body composition, waist circumference has been chosen for its simple, fast, and cheap measurement in population-based studies. Several studies have reported a high correlation between waist circumference and metabolic diseases. During the last few years, waist circumference has been encouraged in clinical practice* better to detect subjects at increased risk of metabolic diseases.

2. Please about waist circumference explanation to include in methods and materials part, section 2.3;

3. Please clarify the name of these tables - is it really association with insulin resistanece or HOMA-IR and HOMA AD values?

Table 3. Association between waist circumference and gut microbiota with insulin resistance.

Table 4. Direct, indirect, and total effects of gut microbiota on insulin resistance.

5. Please include this data in results part:

Regarding the HOMA-IR score, 38 children have HOMA-IR >2, and 495 children have HOMA-IR <2, for a total of 533 children.

In Table nr1: Yes or No

Author Response

Dear Reviewer 2

Again, we appreciate the comments and observations; the time invested in reviewing our manuscript is valuable.

We have shared the responses to the comments below.

  1. This citation is from discussion part:

Within the limitations of our study, the food frequency questionnaire (FFQ) is the 347mostfrequentlyused method in population studies as it allows long-term dietary intake 348to be assessed in a relatively simple, cost-effective, and time-efficient way. The diet infor-349mation was obtained throughaself-report. We are aware of participants’ inability to re-350member their intakes accurately and completely; however, we adjusted the models based 351on macronutrient consumption (“all-components model”) to provide a less biased esti-352mate[45,46].In addition, the FFQ was applied in the same way to all the participants in 353the presence of their parents to standardize the information.

R= This paragraph has been removed from the discussion section.

Please change this explanation about study limitation, like you answer me in comments:

“Within the limitations of our study, we evaluated RI using the HOMA index, with the awareness that the hyperinsulinemic–euglycemic clamp (HEC) is the gold standard for assessing insulin sensitivity. The HOMA index imputes the dynamic function of β-cells, which prevents direct measurement of these cells' proper insulin secretion function. Nevertheless, the HOMA index is an inexpensive quantitative tool that works adequately in population studies. The validity between the HOMA index and the HEC in the pediatric population has also been reported, allowing us to use this index as a valuable tool for classification in epidemiological studies.”

“Among anthropometric markers of central adiposity, waist circumference is the best predictor of insulin resistance in children. Although dual-energy X-ray absorptiometry (DXA) is the gold standard for measuring body composition, waist circumference has been chosen for its simple, fast, and cheap measurement in population-based studies. Several studies have reported a high correlation between waist circumference and metabolic diseases. During the last few years, waist circumference has been encouraged in clinical practice to detect subjects at increased risk of metabolic diseases. If there were a measurement error among the anthropometrists, it would be non-differential.”

R= The above paragraphs have been added to the discussion section.

  1. Please about waist circumference explanation to include in methods and materials part, section 2.3;

R= The following statement has been added to the methods section.

“The personnel who performed the measurement were trained and standardized in the correct measurement in children with calibrated instruments.”

  1. Please clarify the name of these tables - is it really association with insulin resistanece or HOMA-IR and HOMA AD values?

R= We agree with the observation, the table names have changed.

Table 3. Association between waist circumference and gut microbiota with HOMA-IR and HOMA-AD.

Table 4. Direct, indirect, and total effects of gut microbiota on HOMA-IR and HOMA-AD.

Table 5. Direct, indirect, and total effects of gut microbiota profiles on HOMA-IR and HOMA-AD.

  1. Please include this data in results part:

Regarding the HOMA-IR score, 38 children have HOMA-IR >2, and 495 children have HOMA-IR <2, for a total of 533 children.

R= Because this information is not shown in Table 1, we decided to place the following statement at the end of the description of the results in Table 1.

“The prevalence of children with HOMA-IR >2 was 7% (n=38), and children with HOMA-IR <2 was 93% (n= 495).”

  1. Comments on the Quality of English Language

In Table nr1: Yes or No.

R= The change has been made. The manuscript was sent to MDPI for English editing with ID English-69459. You will find the certificate in the attached documents.

This manuscript is a resubmission of an earlier submission. The following is a list of the peer review reports and author responses from that submission.

Round 1

Reviewer 1 Report

I would suggest that the authors, however, seriously consider the design of the study. The research group is large enough, 533 children, to be divided into 3 groups - children with normal weight, overweight and obesity.

The age of the children should also be taken into account, because a 6-year-old child is in prepuberty, while a 12-year-old child may already be in the puberty period, when physiological insulin resistance is typical.

I also did not understand the chosen value of waist circumference > 63.6cm. I would also suggest the waist circumference evaluate by charts (are or not central obesity).

Fig1 is very complex and difficult to understand, however, it is necessary to simplify this picture.

A lot of displayed data that are not related to the main research question - waist circumference, insulin resistance and microbiota. That for example Physical activity (METs/hour/week) etc.

In the discussion, more attention would be paid to the discussion of a still researched question - obesity, insulin resistance and the role of microbiota in children!

The quality of the English language should be significantly improved, the meaning of the obtained results should be described very precisely in the results section.

Author Response

We thank you for your observations and comments on our work. We are sending the response to the concerns of the manuscript. 

Reviewer 1

  1. I would suggest that the authors, however, seriously consider the design of the study. The research group is large enough, 533 children, to be divided into 3 groups - children with normal weight, overweight and obesity.

We appreciate the observation; however, when performing the power calculation for mediation analysis with the AM Schoemann methodology for indirect effects, a sample of 310 participants must have sufficient power to detect effects.

Our total sample (N=533) of children is composed of the following:

normal weight (265)

overweight (103)

obesity (165)

Under our hypothesis where waist circumference participates as a mediator, we would have an overfit in our statistical models if we stratify by BMI Z-scores.

Schoemann AM, Boulton AJ, Short SD. “Determining Power and Sample Size for Simple and Complex Mediation Models”. Soc Psychol Personal Sci. 2017, 8(4):379–86.

Schisterman EF, Cole SR, Platt RW. Overadjustment bias and unnecessary adjustment in epidemiologic studies. Epidemiology. 2009 Jul;20(4):488-95. 

  1. The age of the children should also be taken into account, because a 6-year-old child is in prepuberty, while a 12-year-old child may already be in the puberty period, when physiological insulin resistance is typical.

We fully agree that age is a confounder in our association because it is associated with gut microbiota, is associated with insulin resistance, and is not part of the causal chain. Because we may have spurious associations, we decided to adjust for age in the statistical models to account for the effect of age.

  1. I also did not understand the chosen value of waist circumference > 63.6cm. I would also suggest the waist circumference evaluate by charts (are or not central obesity).

The WHO has not established values for waist circumference in children as it has done with BMI Z-scores for age and sex, according to Yamanaka, determined the cut-off points for waist circumference in boys 6-8 years (78th percentile, 63.58 cm) and girls of the same age (80th percentile, 63.63 cm). Fredriksen reported cut-off points for boys aged 6-12 years (90th percentile, 63.0 cm) and girls of the same age (66.0 cm). To adequately represent our study population, we decided to use the value 63.6 cm, which represents the median of our population, in addition to being a value very close to the cut-off points reported by other authors.

Yamanaka, Ashley B et al. “Determination of Child Waist Circumference Cut Points for Metabolic Risk Based on Acanthosis Nigricans, the Children's Healthy Living Program.” Preventing chronic disease vol. 18 E64. 24 Jun. 2021.

Fredriksen, Per Morten et al. “Waist circumference in 6-12-year-old children: The Health Oriented Pedagogical Project (HOPP).” Scandinavian journal of public health vol. 46,21_suppl (2018): 12-20.

  1. Fig1 is very complex and difficult to understand, however, it is necessary to simplify this picture.

We agree on the complexity of the DAG; however, according to MA Hernán, the DAG may appear complex as all assumptions and expert knowledge must be captured in this diagram. This diagram allows us to summarize the knowledge and assumptions intuitively, which will help us communicate our results effectively. The figure cannot be simplified as all possible relationships between our variables must be captured.

The guide for reporting mediation analysis indicates the presence of this diagram; the entire manuscript is based on this guide (AGReMA) for the communication of results.

Hernán MA, Robins JM (2020). Causal Inference: What If. Boca Raton: Chapman & Hall/CRC.

Lee, Hopin et al. “A Guideline for Reporting Mediation Analyses of Randomized Trials and Observational Studies: The AGReMA Statement.” JAMA vol. 326,11 (2021): 1045-1056.

  1. A lot of displayed data that are not related to the main research question - waist circumference, insulin resistance and microbiota. That for example Physical activity (METs/hour/week) etc.

We present the information on physical activity and family history of DT2 in Table 1; both variables were detected as potential confounders in the DAG. We included both variables to adjust the statistical models; however, they did not contribute variability to the estimator. In the statistical analysis section, we report the decision to exclude physical activity and family history of DT2 as adjustment variables.

  1. In the discussion, more attention would be paid to the discussion of a still researched question - obesity, insulin resistance and the role of microbiota in children!

We appreciate the suggestion. We have added this information to the discussion section. The grammar has been reviewed, and the results have been clearly described.

Reviewer 2 Report

Carlos et al performed a cross-sectional study on the relationship between relative abundance of five gut microbes, waist circumference and insulin resistance in 533 children. While the effort was appreciated, I have the following concerns that I would like to discuss with the authors.

1. This was a cross-sectional study, and causality could not be easily established. This challenges the proposed DAG (and the mediation analysis). For example, although waist circumference may be a good, non-invasive predictor of insulin resistance (IR), IR might partly induce increased waist circumference (as discussed in 10.1186/1472-6823-9-1).

2. The observed correlations may be attributed to age difference. It was not clearly stated how the BMI z-scores were calculated. Were they age-specific? Were the waist circumferences normalized in age strata? According to Table 1, those who were overweight or obese were significantly older compared to their normal-weight counterparts. Therefore, the gut microbes might just be “bystanders” of the correlations between age and waist circumferences/IR: they simply changed with age but were not a cause or a consequence of the phenotype.

3. It was not clear why the five microbes were targeted. The Introduction stated some general knowledge of microbiota, but did not narrow down the scope to explain what was the research gap/preliminary results of the current study. The author described the aim at the end of the section, all of a sudden.

4. The authors confused (a few) microbes and “microbiota”, the latter being excessively and wrongly used throughout the manuscript.

Minor concerns:

You may want to spell numbers below ten (e.g. 5 -> five). There is a typo of “and”, as “anb”.

Author Response

We thank you for your observations and comments on our work. We are sending the response to the concerns of the manuscript. 

Reviewer 2

  1. This was a cross-sectional study, and causality could not be easily established. This challenges the proposed DAG (and the mediation analysis). For example, although waist circumference may be a good, non-invasive predictor of insulin resistance (IR), IR might partly induce increased waist circumference (as discussed in 10.1186/1472-6823-9-1).

We agree that cross-sectional studies do not allow us to identify causal effects because the exposure and event are measured simultaneously; however, it has been encouraged to have better "causal" questions with observational designs (including cross-sectional designs). We have not used causal language to describe our results, but our question and objective stem from a hypothesis that suggests causality. Hamaker states that cross-sectional studies can evaluate "causal hypotheses" if the results are discussed cautiously. As for statistical analysis, path analysis and structural equation modeling do not allow us to evaluate causality; however, we can evaluate theoretical causal hypotheses. In prospective studies, as discussed in the limitations section, it has been described that changes in microbiota precede different adiposity phenotypes.

Hernán MA. The C-Word: Scientific Euphemisms Do Not Improve Causal Inference From Observational Data. Am J Public Health. 2018 May;108(5):616-619. 

Hamaker EL, Mulder JD, van IJzendoorn MH. Description, prediction and causation: Methodological challenges of studying child and adolescent development. Dev Cogn Neurosci. 2020 Dec;46:100867.

  1. The observed correlations may be attributed to age difference. It was not clearly stated how the BMI z-scores were calculated. Were they age-specific? Were the waist circumferences normalized in age strata? According to Table 1, those who were overweight or obese were significantly older compared to their normal-weight counterparts. Therefore, the gut microbes might just be “bystanders” of the correlations between age and waist circumferences/IR: they simply changed with age but were not a cause or a consequence of the phenotype.

The statistical analysis section will find that the BMI Z-scores are for age according to the values described by WHO.

We appreciate the observation; if we normalize waist circumferences according to age strata at the time of inferential analysis, we will cause overfitting in the models because waist circumference (collinearity with BMI Z-scores) is mediating our hypothesis.

The WHO has not established values for waist circumference in children as it has done with BMI Z-scores for age and sex, according to Yamanaka, determined the cut-off points for waist circumference in boys 6-8 years (78th percentile, 63.58 cm) and girls of the same age (80th percentile, 63.63 cm). Fredriksen reported cut-off points for boys aged 6-12 years (90th percentile, 63.0 cm) and girls of the same age (66.0 cm). To adequately represent our study population, we decided to use the value 63.6 cm, which represents the median of our population, in addition to being a value very close to the cut-off points reported by other authors.

We fully agree that age is a confounder in our association because it is associated with gut microbiota, is associated with insulin resistance, and is not part of the causal chain. Because we may have spurious associations, we decided to adjust for age in the statistical models to account for the effect of age.

Yamanaka, Ashley B et al. “Determination of Child Waist Circumference Cut Points for Metabolic Risk Based on Acanthosis Nigricans, the Children's Healthy Living Program.” Preventing chronic disease vol. 18 E64. 24 Jun. 2021.

Fredriksen, Per Morten et al. “Waist circumference in 6-12-year-old children: The Health Oriented Pedagogical Project (HOPP).” Scandinavian journal of public health vol. 46,21_suppl (2018): 12-20.

  1. It was not clear why the five microbes were targeted. The Introduction stated some general knowledge of microbiota, but did not narrow down the scope to explain what was the research gap/preliminary results of the current study. The author described the aim at the end of the section, all of a sudden.

These five species are chosen because they have gained great relevance within the international literature. Likewise, within our work group, we have detected that these five bacteria are associated with metabolic diseases in children.

Orbe-Orihuela, Yaneth C, Lagunas-Martínez, Alfredo, Bahena-Román, Margarita, Madrid-Marina, Vicente, Torres-Poveda, Kirvis, Flores-Alfaro, Eugenia, Méndez-Padrón, Araceli, Díaz-Benítez, Cinthya E, Peralta-Zaragoza, Oscar, Antúnez-Ortiz, Diana, Cruz, Miguel, & Burguete-García, Ana I. (2018). High relative abundance of firmicutes and increased TNF-α levels correlate with obesity in children. Salud Pública de México60(1), 5-11. 

Estrada-Velasco, Barbara Ixchel, Cruz, Miguel, García-Mena, Jaime, Valladares Salgado, Adan, Peralta Romero, Jesus, Guna Serrano, Maria de los Remedios, Madrid-Marina, Vicente, Orbe Orihuela, Citlalli, López Islas, Claudia, & Burguete-García, Ana Isabel. (2015). La obesidad infantil como consecuencia de la interacción entre firmicutes y el consumo de alimentos con alto contenido energético. Nutrición Hospitalaria31(3), 1074-1081.

Huerta-Ávila, E. E., Ramírez-Silva, I., Torres-Sánchez, L. E., Díaz-Benítez, C. E., Orbe-Orihuela, Y. C., Lagunas-Martínez, A., Galván-Portillo, M., Flores, M., Cruz, M., & Burguete-García, A. I. (2019). High Relative Abundance of Lactobacillus reuteri and Fructose Intake are Associated with Adiposity and Cardiometabolic Risk Factors in Children from Mexico City. Nutrients11(6), 1207.

Castañeda-Márquez, A. C., Díaz-Benítez, C. E., Bahena-Roman, M., Campuzano-Benítez, G. E., Galván-Portillo, M., Campuzano-Rincón, J. C., Lagunas-Martínez, A., Bermudez-Morales, V. H., Orbe-Orihuela, Y. C., Peralta-Romero, J., Cruz, M., & Burguete-García, A. I. (2020). Lactobacillus paracasei as a protective factor of obesity induced by an unhealthy diet in children. Obesity research & clinical practice14(3), 271–278.

Ayala-García, J. C., Lagunas-Martínez, A., Díaz-Benítez, C. E., Orbe-Orihuela, Y. C., Castañeda-Márquez, A. C., Ortiz-Panozo, E., Bermúdez-Morales, V. H., Bahena-Román, M., Cruz, M., & Burguete-García, A. I. (2022). High Relative Abundance of Staphylococcus aureus and Serum Cytokines Are Associated with Cardiometabolic Abnormalities in Children. Metabolic syndrome and related disorders20(5), 303–311.

  1. The authors confused (a few) microbes and “microbiota”, the latter being excessively and wrongly used throughout the manuscript.

 We appreciate the comment; however, there is no confusion between the terms. The gut microbiota is composed of different microorganisms, mainly bacteria. We measure bacteria in feces; this group of bacteria is part of the gut microbiota.

Minor concerns:

You may want to spell numbers below ten (e.g. 5 -> five). There is a typo of “and”, as “anb”.

The observations have been corrected.

The grammar has been reviewed, and the results have been clearly described.

Reviewer 3 Report

The manuscript adresses a novel topic and has a strong statistical analysis beneath, being conducted on an impressive number of children. Substantial revision is required, while taking into consideration the following comments:

1. Each abbreviation should be defined first time it appears in the text/abstract (please define HOMA-IR and HOMA-AD within the abstract).

2. One could drop the headings (background, etc) from the abstract and rephrase the conclusion, which does not clearly explain the relationship between waist circumference, RA of S. aureus and A. muciniphila and insulin resistance.

3. The introduction contains almost no data regarding the relationship between insulin resistance and imbalance of gut microbiota.

4. "Our aim was to evaluate the direct, indirect (mediated by waist circumference), and total effects on the association between gut microbiota and insulin resistance in Mexican children." Please rephrase the objective, as it is unclear ("effects on the association")

5. Lines 69-73 should be moved to the statistical analysis chapter.

6. I would prefer if the authors represented continuous variables as mean +/- SDs.

7. The statistical analysis chapter contains no data regarding the type of normality test applied to assess Gaussian/non-Gaussian distribution of the analyzed data. These results drive the type of test which is performed for mean/median comparison.

8. English language requires significant revision, maybe performed by a native speaker/language expert.

English language requires significant revision, maybe performed by a native speaker/language expert.

Author Response

We thank you for your observations and comments on our work. We are sending the response to the concerns of the manuscript. 

Reviewer 3

  1. Each abbreviation should be defined first time it appears in the text/abstract (please define HOMA-IR and HOMA-AD within the abstract).

The observations have been corrected.

  1. One could drop the headings (background, etc) from the abstract and rephrase the conclusion, which does not clearly explain the relationship between waist circumference, RA of S. aureus and A. muciniphila and insulin resistance.

The conclusion has been rewritten.

  1. The introduction contains almost no data regarding the relationship between insulin resistance and imbalance of gut microbiota.

We have added this information to the introduction section.

  1. "Our aim was to evaluate the direct, indirect (mediated by waist circumference), and total effects on the association between gut microbiota and insulin resistance in Mexican children." Please rephrase the objective, as it is unclear ("effects on the association")

We have reformulated the aim: This study evaluated the mediating role of waist circumference in the association between GM and insulin resistance (IR) in children.

  1. Lines 69-73 should be moved to the statistical analysis chapter.

We appreciate the comment; however, according to the AGReMa guidelines for reporting mediation analysis, we are instructed to report the sample size calculation in this section after the participants.

Lee, Hopin et al. “A Guideline for Reporting Mediation Analyses of Randomized Trials and Observational Studies: The AGReMA Statement.” JAMA vol. 326,11 (2021): 1045-1056.

  1. I would prefer if the authors represented continuous variables as mean +/- SDs.

Our variables do not follow a normal distribution, so we have presented the median and the 25th and 75th percentiles.

  1. The statistical analysis chapter contains no data regarding the type of normality test applied to assess Gaussian/non-Gaussian distribution of the analyzed data. These results drive the type of test which is performed for mean/median comparison.
  2.  

We have added the following statement to our statistical analysis section:

We performed the Shapiro-Wilk test to evaluate normality. Because the distribution of the variables was not normal, a comparison between groups was performed using the Mann-Whitney U test for continuous variables and chi-square (X2) for categorical variables.

  1. English language requires significant revision, maybe performed by a native speaker/language expert.

The grammar has been reviewed.